# Antimicrobial proteins from oyster hemolymph improve the efficacy of conventional antibiotics

Kate Summer[1]*, Qi Guo[1], Lei Liu[1], Bronwyn Barkla[1], Sarah Giles[2], Kirsten Benkendorff[3]

1 Faculty of Science and Engineering, Southern Cross University, Lismore, NSW, Australia, 2 Flinders Accelerator for Microbiome Exploration, College of Science and Engineering, Flinders University, Bedford Park, SA, Australia, 3 National Marine Science Centre, Southern Cross University, Coffs Harbour, NSW, Australia

* k.summer@outlook.com.au

**Data Availability Statement:** All relevant data are within the paper and its Supporting Information files.

## Abstract

Discovering new antibiotics and increasing the efficacy of existing antibiotics are priorities to address antimicrobial resistance. Antimicrobial proteins and peptides (AMPPs) are considered among the most promising antibiotic alternatives and complementary therapies. Here, we build upon previous work investigating the antibacterial activity of a semi-purified hemolymph protein extract (HPE) of the Australian oyster *Saccostrea glomerata*. HPE showed antimicrobial-biofilm inhibitory activity toward laboratory and clinical strains of *Streptococcus pneumoniae* and *Streptococcus pyogenes* at 4.4 and 24.1 µg/mL total protein, respectively. In combination assays, the effectiveness of conventional antibiotics (ampicillin, gentamicin, trimethoprim and ciprofloxacin) was improved between 2 to 32-fold in the presence of HPE (1–12 µg/mL) against a range of clinically important bacteria including *Streptococcus* spp., *Pseudomonas aeruginosa*, *Moraxella catarrhalis*, *Klebsiella pneumoniae* and *Staphylococcus aureus*. Effective HPE concentrations are comparable to AMPPs currently approved for use or in clinical trials pipelines. Proteomics analysis of HPE identified a number of proteins including abundant known AMPPs. It was non-toxic to A549 human lung cells up to 205 µg/mL, demonstrating safety well above effective concentrations. Activity was retained with storage at -80°C and ambient laboratory temperature (~24°C), but declined after treatment at either 37°C or 60°C (1 h). This study is in agreement with growing evidence that AMPPs show specificity and a high capacity for synergism with antibiotics. The discovery of HPE provides great opportunities for both pharmaceutical and aquaculture industry development.

## 1. Introduction

The discovery and development of new antibiotics is a global public health priority. Bacterial infections are a leading cause of disease and treatment is reliant on antibiotics [1]. The rapid development of antimicrobial resistance (AMR) is diminishing the effectiveness of available

**Funding:** Financial support was provided by the Australian Government Department of Education Research Training Program, the Faculty of Science and Engineering at Southern Cross University, and the Malacological Society of Australasia. Proteomics analysis was made possible by ARC LIEF funding (LE170100192). The funders had no role in study design, data collection and analysis, decision to publish, or preparation of the manuscript.

**Competing interests:** The authors have no competing interests to declare.

front-line antibiotics, whilst there are major shortfalls in the production of new drugs [1]. Drug-resistant bacteria are responsible for increasing rates of untreatable infections such that AMR is now associated with around five million deaths each year [1, 2]. We also now know that biofilms are a feature of most infections, representing a protected mode of bacterial growth that affords tolerance to high antibiotic concentrations [3]. Taken together, there is a critical need to discover new compounds with novel mechanisms of action (i.e., structurally different to existing classes of antibiotics, able to prevent/disperse biofilms) to preserve the ability to prevent and treat infectious disease and to reduce overexposures to existing antibiotics that promote development of resistance [4].

Antimicrobial proteins and peptides (AMPPs) are a diverse group of small proteins forming part of the humoral immune systems of most multicellular organisms [5, 6]. There is thus a plethora of unconventional sources of AMPPs with unique activities [5, 7, 8]. AMPPs have been classically assumed to possess broad-spectrum activity and simple kinetics, but recent evidence suggests high specificity and capacity for synergism (i.e., where the effect of a combination therapy is greater than the sum of each component) [7, 9]. Synergies between AMPPs and antibiotics present exciting possibilities for future use of combination therapies in clinical settings [10–15]. Most cases of synergistic interaction have been observed between AMPPs which affect the permeability of bacterial cell membranes, and antibiotics affecting the biosynthesis of nucleic acids and proteins which must penetrate the cell in order to exert their bactericidal activity [16]. Further benefits include generally low toxicity and unlikely development of AMR [9]. Consequently, AMPPs are considered among the most promising pharmacological leads [6, 17]. There are currently five AMPPs (four bacteria-derived and one invertebrate-derived) in clinical use as alternative antibiotics whilst at least another 35 are under clinical evaluation [18]. AMPPs are also produced by humans but, if isolated as treatments and resistance should develop, there is a risk of collateral development of resistance to endogenous immunity, so non-human sources are preferred [7]. AMPPs from marine invertebrates, particularly the Mollusca, which have evolved strong innate immunity to compensate for a lack of acquired immunity, are receiving notable attention in this field [11, 19–29].

We recently tested a semi-purified HPLC-fractionated hemolymph protein extract (HPE) from the Sydney Rock Oyster (SRO), *Saccostrea glomerata*, which showed significant bactericidal/biofilm inhibitory activity against *Streptococcus pneumoniae* [30]. Here, we sought to screen a broader range of bacterial pathogens, with a particular focus on those causing respiratory infections. Combination experiments were carried out to assess synergistic activity between HPE and conventional antibiotics. We also investigated cytotoxicity and thermal stability to establish clinical relevance.

## 2. Materials and methods

### 2.1 HPE preparation

**2.1.1 Hemolymph collection from SRO.** Live SRO were sourced from the Clyde River, NSW, Australia via a commercial supplier. Oysters were shucked and the pericardial region was immediately punctured using a sterile syringe and 26-gauge needle [30, 31]. Hemolymph was withdrawn from multiple oysters (average 16 individuals per pool), combined into 5 mL pools which were maintained on ice, and filtered to 0.2 μM to obtain cell-free hemolymph. Samples were frozen at -80˚C then freeze dried over 24 h (Christ Alpha 1–4 LD plus, at -55˚C and vacuum sealed to 0.35 mbar). Freeze-dried powder in each pool was resolubilised in MilliQ water, resulting in hemolymph that was 5 times more concentrated than in the original organism.

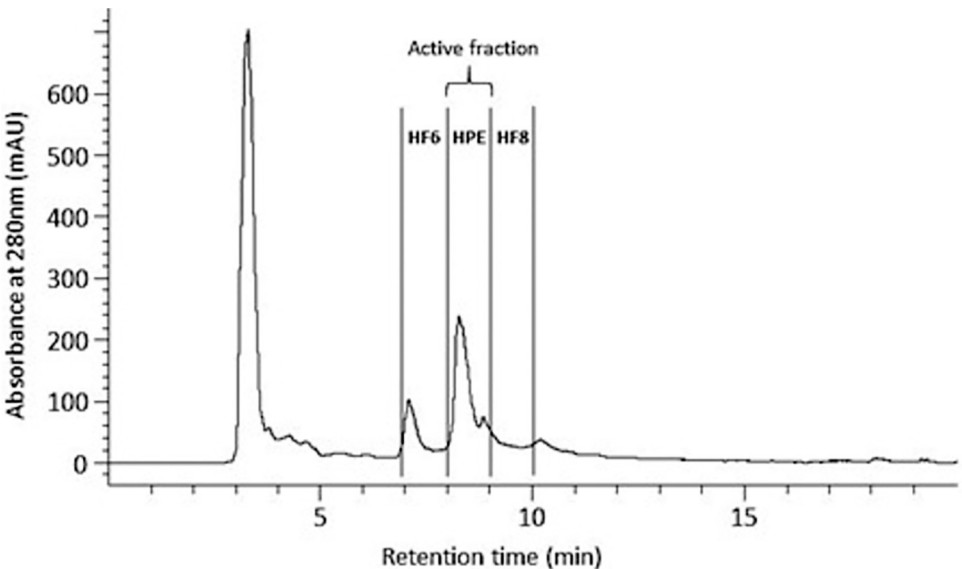

**Fig 1. Analysis of a typical SRO cell-free hemolymph sample obtained by preparative HPLC.** The chromatogram at 280 nm (specific to proteins [54]) shows the relative composition of hemolymph components across the three fractions collected between 7–10 mins at 1 min time slices, where HPE showed strong antibacterial-antibiofilm activity and HF6 and HF8 were relatively inactive. The full UV spectra are supplied in S2 Fig in S2 File.

**2.1.2 Hemolymph fractionation.** Fractionation of concentrated cell-free hemolymph was carried out using an Agilent 1260 Infinity Prep high-performance liquid chromatography (HPLC) system equipped with a Phenomenex Jupiter 5u C18, 250 x 10 mm, 300Å column [30]. Absorbance was monitored at 210 and 280 nm. The data was collected using ChemStation B.04.03. The mobile phase included acetonitrile (ACN) with 0.05% trifluoroacetic acid (TFA), and Milli-Q water with 0.05% TFA. The elution gradient commenced at 5% ACN, increased to 99% at 15 min, and returned to 5% at 16.5 min and was maintained at 5% until 20 min for re-equilibration. The flow rate of the mobile phase was 5 mL/min. The sample injection volume was 400 µL with five injections per run (15 injections were carried out in total). Three fractions were collected at 1 min time-slices between 7–10 min (Fig 1). HPE was comparable to the active Fraction 7 in [30]. The two fractions either side of HPE were named HF6 and HF8 (comparable to less active fractions 6 and 8 in [30]) (Fig 1). Fractions were freeze dried at -80˚C (Christ Alpha LOC-1M) over two days then vacuum dried for a further two days. Each fraction was resolubilised to 500 µL in sterile phosphate buffered saline (PBS) and aliquoted into Eppendorf tubes, then stored at -80˚C for use in respective assays/analyses.

**2.1.3 HPE heat treatments.** Subsamples of HPE fractions stored at -80˚C were compared to: i) those stored at ambient laboratory temperature (around 24˚C) for 24 hours; ii) heated in a water bath at 37˚C for one hour; iii) heated in a water bath at 60˚C for one hour. These temperatures and exposure durations were selected to represent conditions that may be encountered during experimentation or administration e.g., thawed/laboratory temperature, body temperature, overheating during transport. Changes in the HPE proteome and antibacterial-biofilm inhibitory activity against *S. pneumoniae* (ATCC 51916) were evaluated as per methods described below.

## 2.2 Proteomics

**2.2.1 Protein quantification (Bradford assay) and visualisation (SDS-PAGE).** Total protein concentration was determined in 96-well plates according to the Bradford method

[32] with Bovine Serum Albumin (BSA; lyophilised powder, CAS: 9048-46-8, Sigma-Aldrich) as the standard. HPE samples were diluted 1:1 in sterile PBS (Sigma Aldrich) to a total volume of 10 μL, then 100 μL protein dye reagent (Bio-Rad, Australia) was added to each well [30]. Absorbance was measured spectrophotometrically at OD 595. All standards and samples were prepared in triplicate and protein concentrations were estimated according to the BSA standard curve.

Proteins in HPE (and the two fractions collected either side) were separated using pre-cast polyacrylamide gels (Bio-Rad Mini-PROTEAN TGX, 10-well, 12% single percentage gel). A sample representing 1–2 μg total protein was prepared by mixing 3.5 μL sample with 7 μL 2x Laemmli sample buffer (BioRad, Australia). Samples were warmed only to 30°C for 2 minutes. Ten μL of samples were added to respective lanes, alongside 10 μL molecular weight marker (Precision Plus Protein™ Dual Xtra pre-stained protein standard, BioRad, Australia) which was positioned in the first well of each gel. and 10 μL Laemmli buffer was added to remaining empty wells. The gels were electrophoresed at 140 V in running buffer solution (25mM Tris, 192 mM glycine, 0.1% SDS; Bio-Rad, Australia) until the sample reached the end of the gel. Gels were placed in fixing solution (40% v/v methanol and 7% v/v acetic acid in Milli-Q) for 40 min, then rinsed and placed in Coomassie blue staining solution (2.5 g Coomassie brilliant blue R250 dissolved in 50% v/v methanol, 7% v/v acetic acid in Milli Q) overnight with gentle agitation. Gels were then de-stained with a solution of 50% (v/v) methanol and 10% (v/v) acetic acid in Milli-Q water with gentle agitation over 3 h. Finally, the gel was photographed on a white light illuminating box.

**2.2.2 Protein identification by HPLC-MS/MS.** Proteins in HPE, as well as HF6 and HF8, were identified by microflow HPLC-mass spectrometry/mass spectrometry (HPLC-MS/MS) on an Eksigent, Ekspert nanoLC 400 system (SCIEX, Canada) coupled to a Triple Time of Flight (TOF) 6600 mass spectrometer (SCIEX, Canada) equipped with a micro-Duo Spray ion source, as per [30]. Subsamples of HPE, HF6 and HF8, were trypsin digested first at 5°C for 30 min and then at 37°C overnight, and the resulting peptides were recovered by three extractions with 35 μL of 50% (v/v) acetonitrile with 2% (v/v) formic acid. The extracts were dried in a vacuum centrifuge and redissolved in 15 μL of 5% formic acid before being analyzed. A volume of 5 μl from each extract was injected onto a 5 mm x 300 μm, C18, 3 μm trap column (SGE, Australia) for 6 min at 10 μL/min. The trapped extracts were then washed onto the analytical 300 μm x 150 mm Zorbax 300SB-C18 3.5 μm column (Agilent Technologies, USA) at a flow rate of 3 μL/min and a column temperature of 45°C. Solvents for peptide elution were: A) 0.1% formic acid in MilliQ water and B) 0.1% formic acid in ACN. Linear gradients of 2–25% solvent B over 60 min at 3 μL/minute flow rate, followed by a steeper gradient from 25% to 35% solvent B in 13 min, then 35% to 80% solvent B in 2 min. The gradient was then returned to 2% solvent B for equilibration prior to the next injection. The micro ion spray voltage was set to 5500V, de-clustering potential (DP) 80V, curtain gas flow 25, nebulizer gas 1 (GS1) 15, heater gas 2 (GS2) 30 and interface heater at 150°C. The mass spectrometer acquired 250ms full scan TOF-MS data followed by 50ms full scan product ion data, with a rolling collision energy, in an Information Dependent Acquisition (IDA) scan mode. Full scan TOF-MS data was acquired over the mass range m/z 350–2000 and for product ion ms/ms, m/z 100–1500. Ions observed in the scan exceeding a threshold of 150 counts and a charge state of +2 to +5 were set to trigger the acquisition of product ion, ms/ms spectra of the resultant 30 most intense ions. The data was acquired and processed using Analyst TF 1.7 software (ABSCIEX, Canada).

**2.2.3 Protein data analysis.** Protein Pilot 5.0.2 (SCIEX, Canada) was used to search spectra against the UniProt Mollusca database (1 186 286 entries, 2024) [33]. Scaffold 4.8.6 (Proteome Software, USA) was used to validate MS/MS-based protein identification and

quantification [30]. Normalised spectral abundance factor (NSAF) was used for quantification and comparisons between samples [34]. NSAF normalises spectral counts by protein length, ensuring accurate relative abundance measurements. This method is simple and robust, and its log-transformed values facilitate statistical analyses, allowing for effective cross-sample comparisons [34, 35]. Principal Component Analysis (PCA) [36] and hierarchical clustering (Ward's method) [37] were undertaken in R (4.1.0) [38] to visualise the abundance of unique proteins in HPE. These clustering methods are generally applied to analyse and visualise the relationships and differences between samples based on the quantified data. For proteins upregulated in HPE, we collated functional annotations as listed in the UniProt database (https://www.uniprot.org/; accessed February, 2024) and searched the published literature for relevance to antimicrobial activity.

## 2.3 Antibacterial and antibiofilm assays

**2.3.1 Media and reagents.** All reagents used were analytical (HPLC) grade purchased from Sigma Aldrich unless specified otherwise. Media and inoculum were prepared according to species-specific requirements (S1 Table in S1 File).

**2.3.2. Bacteria preparation.** Laboratory strains of bacteria used in this study were: *S. pneumoniae* (ATCC 51916), *Streptococcus pyogenes* (ATCC 19615), *Staphylococcus aureus* (ATCC 25923), *Klebsiella pneumoniae* (ATCC 13883), and non-typeable *Haemophilus influenzae* (ATCC 10211). Clinical strains included *S. pneumoniae* (serotypes 19F, 6B and 14) and *Moraxella catarrhalis* (K65, non-clumping variant), which were originally isolated from a patient with chronic otitis media, and *Pseudomonas aeruginosa* (serotype 385), originally isolated from a chronically infected patient with cystic fibrosis (S1 Table in S1 File). The *S. pneumoniae* (ATCC 51916) strain, for which we have previously established susceptibility to HPE [30], was used as a reference. These species were selected as they are among the leading causes of global bacterial infections, often manifesting with reduced susceptibility to multiple classes of antibiotics [1, 39, 40]. They are especially implicated in respiratory infections [40], as well as meningitis, sepsis, and skin/wound infections [39, 41–43]. The specific strains and isolates are recommended for quality control and/or are highly representative of clinical infections (S1 Table in S1 File).

Cryopreserved bacteria were revived on agar and grown to log-phase over 20–22 h at 37˚C with 5% $CO_2$. Isolated colonies were suspended in 1 mL media and grown to log-phase in a shaking incubator at 37˚C with 5% $CO_2$ for 1–5 h (S1 Table in S1 File) until blank-corrected absorbance was 0.1–0.2, as measured spectrophotometrically at 600 nm (BioRad iMark™ microplate reader), equivalent to ~$10^8$ colony forming units per ml (CFU/mL). Cultures were diluted in media to achieve a working suspension of $10^6$ CFU/mL, finally reduced to $5{\times}10^5$ CFU/mL in assays. CFU's were confirmed by plating dilutions of working suspensions used in each assay.

**2.3.3 Antibacterial-biofilm inhibition coupled assays.** Antibacterial activity was determined using the liquid growth microdilution assay according to standard screening procedures [44] in 96-well plates with subsequent crystal violet staining and determination of biofilm inhibition [45]. HPE was screened for activity against the suite of bacterial pathogens. Heat treated HPE, HF6 and HF8 were also screened against *S. pneumoniae* ATCC 51916 for reference. When reconstituted to between 0.3–0.5 mL in PBS then diluted in assays, the highest total protein concentration of HPE was 150.7 μg/mL (± 5.1 SD), which was serially diluted ten-fold. The top concentrations of HF6 and HF8 were 100.5 μg/mL (± 5.3 SD) and 137.8 μg/mL (± 1.0 SD).

HPE was also tested in combination with antibiotic standards at concentrations below minimum inhibitory concentrations (MICs) for each component (S1 Table in S1 File). Top

concentrations of antibiotics started at 2x MICs and were diluted ten-fold. For *S. pneumoniae* (all strains) and *S. pyogenes*, ampicillin was combined with 1 and 3 μg/mL HPE. For *P. aeruginosa*, Nt*Hi*, *M. catarrhalis*, *S. aureus* and *K. pneumoniae*, control antibiotics (gentamicin, ciprofloxacin, ampicillin and trimethoprim, respectively) were combined with 6 or 12 μg/mL HPE.

All plates included duplicate bacteria in media controls (normalised to 100% growth), blank media-only negative controls, and serial dilutions of antibiotic standards as positive controls. Plates were incubated for 20–22 h at 37˚C with 5% $CO_2$ then read spectrophotometrically at OD 595 for determination of antibacterial activity (planktonic growth inhibition). The same plates were evaluated for inhibition of biofilm formation by aspirating planktonic cells and media from the wells and rinsing twice with PBS. Remaining biofilms were sprayed liberally with 80% v/v ethanol and allowed to dry, then stained with 200 μL 0.1% crystal violet. After 20 min, excess stain was discarded and plates were again twice-rinsed with PBS. Stained biofilms were solubilized with 200 μL 30% v/v glacial acetic acid and OD was measured at 595 nm (after 5 seconds shaking).

MICs were the minimum concentrations inhibiting growth relative to untreated (media-only) blanks (i.e., treatment absorbance ≤ blank absorbance). Solutions from the MIC well (and 2-fold either side) were diluted 1:10 in sterile PBS, plated to identify presence/absence of growth after overnight incubation to determine minimum bactericidal concentrations (MBC). All raw data were blank corrected. Absorbance measurements from duplicate treatments on each plate were averaged and data from $n$ = 3–5 replicate experiments per bacteria species were used in the analysis. Data are reported as means ± standard deviation (SD). Inhibition of planktonic growth and inhibition of biofilm formation was calculated as percentages relative to respective positive-growth (normalized to 100% growth) controls: % inhibition = 100-([treatment-blank]/[positive growth control-blank])×100.

**2.3.4 Biofilm treatment assays.** *S. pneumoniae* (ATCC 51916) inoculum at $5×10^5$ CFU/mL was incubated in 96-well plates, with 100 μL inoculum per well. After 20–22 h, planktonic cells and media were removed and each well was carefully rinsed twice with 200 μL PBS. Six HPE treatments were prepared separately ranging from 37.5–1.2 μg/mL protein in media, then 100 uL treatments were added to duplicate wells. The treatment concentration range was chosen to incorporate effective concentrations expected on the basis of previous work [30]. Controls were as follows: treatment with 300 μg/mL ampicillin was used as the treatment control (i.e., 100% bactericidal) to account for any residual dead cells in the remaining matrix. Media only (in wells with no-pre-formed biofilms) was used as the blank. The positive growth control contained pre-formed biofilms and media only. The positive treatment controls were EDTA (0.25–0.039M) and ciprofloxacin (50–0.39 μg/mL), which are known antibiofilm agents [46, 47]. Plates were incubated again for 20–22 h. Spectrophotometric readings at OD 595 were taken to determine the extent of bacterial regrowth. Solution from the regrowth-MIC well (and the 2 treatment wells either side, i.e., 2 concentrations above and 2 below) was aspirated and wells were rinsed twice with 200 μL PBS. Biofilms were then scraped along the bottom and edges of the well with a pipette tip and 100 μL PBS was mixed in during scraping [48]. The PBS containing biofilms were then aspirated, vortexed and spread across HBA agar. The presence/absence of growth after overnight incubation indicated the minimum HPE concentration needed to kill cells and prevent regrowth from biofilms.

**2.3.5 Antibacterial data analysis.** We used Stan [49] to fit five-parameter log-logistic dose-response models [50, 51] which were applied to each dataset in R (v. 4.2.1) [38]. We modelled the mean observed response (% inhibition of planktonic growth inhibition and % inhibition of biofilm formation) [30]. Posterior distributions of parameters with medians (i.e., median effective concentration values, $EC_{50}$) and 95% highest posterior density intervals

(HPDI) were summarised. Significance ($p < 0.05$) was assessed based on whether zero was contained within the 95% credible intervals (CI's). Differences in effective concentrations between HPE heat treatments were tested for significance ($p < 0.05$) by analysis of variance (ANOVA) with post-hoc least significant difference (LSD) tests in SPSS (v. 29.0).

## 2.4 Cytotoxicity assays

**2.4.1 A549 human lung cell culture.** The A549 hypotriploid alveolar basal epithelial cell line was used as a model for treatment of respiratory-related pathogens. The A549 cells were grown at 37°C with 5% $CO_2$ in a humidified atmosphere in Dulbecco's Modified Eagle's Medium (DMEM) supplemented with 1 g/L glucose, 10% (v/v) heat inactivated fetal calf serum (FCS) and 1% (v/v) penicillin and streptomycin.

**2.4.2 Preparation of cells for use in cytotoxicity assays.** A549 cells were passaged in T75 culture flasks in a total volume of 20 mL pre-warmed supplemented DMEM and incubated at 37°C in 5% $CO_2$ in a humidified atmosphere for approximately four days or until 70% confluence was achieved. To remove the A549 cells from the flask, DMEM was removed then 2 mL of TrypLE™ Express Enzyme (1X) (no phenol red) (ThermoFisher) was added and the flask was incubated at room temperature for 10 min. Cells were removed and the suspension was centrifuged at 300 x *g* for 5 min, the supernatant was discarded and the pellet was re-suspended in 1mL of fresh DMEM. To prepare for the assay, 1 ml of A549 cells in fresh DMEM was diluted into 19 mL of DMEM (final concentration approximately $4.2 \times 10^5$ cells/mL) then 200µl of DMEM containing A549 cells was added to each experimental well in a 96-well plate.

**2.4.3 Cell viability using the MTS assay.** Fifty microliters of HPE were added to each of the experimental wells at the following final concentrations: 205, 68, 22, 7.6, 2.5, 0.8, 0.28 and 0.09 µg/ml before incubating at 37°C in 5% $CO_2$ for 20 h. The HPE concentration range was selected to be inclusive of the range of tested/effective concentrations used in antimicrobial assays. Negative controls containing 50 µl PBS buffer and untreated cells were also included. Plates contained three replicates of each treatment and the experiment was repeated twice. Cell viability was quantified using an MTS Assay Kit (Cell Proliferation, Colorimetric assay, Abcam, Australia; ab197010) following the manufacturer's protocol. Briefly, 10 µL of MTS reagent (3-(4,5-dimethylthiazol-2-yl)-5-(3-carboxymethoxyphenyl)-2-(4-sulfophenyl)-2H-tetrazolium) was added to cells and cultured at 37°C with 5% $CO_2$ for 2–3 h. Absorbance at 490 nm was determined using an automatic FLUOstar Omega plate reader (BMG Labtech, OMEGA). Differences were tested for significance ($p < 0.05$) by ANOVA, which is robust for cytotoxicity data [52, 53], with post-hoc LSD tests in SPSS (v. 29.0). The data was first tested to ensure the assumptions of normality and homogeneous variances were satisfied for parametric analysis.

## 3. Results

### 3.1 Proteomics

**3.1.1 Hemolymph fractionation and protein quantification.** Concentrated SRO hemolymph contained 1344.3 (±7.0) mg/mL total protein prior to HPLC fractionation. Seven major peaks were detected in chromatograms of the hemolymph. The first large peak (3–4 min) constituted salts, and was discarded (Fig 1). Peaks between 7–10 mins correlated to major protein components (Fig 1). There was no evidence for other potentially-active small molecules based on UV spectra (S2 Fig in S2 File). The average recovery mass of lyophilised powder in HPE was 12.6 (±1.5) mg. The average recovery masses of lyophilised powder in fractions collected before and after HPE were 5.0 (±0.09) mg and 2.6 (±0.7) mg, respectively.

**3.1.2 Protein identification by HPLC-MS/MS.** SDS-PAGE showed differences in the profile and abundances of proteins in collected fractions (S2 Fig in S2 File). The bands around

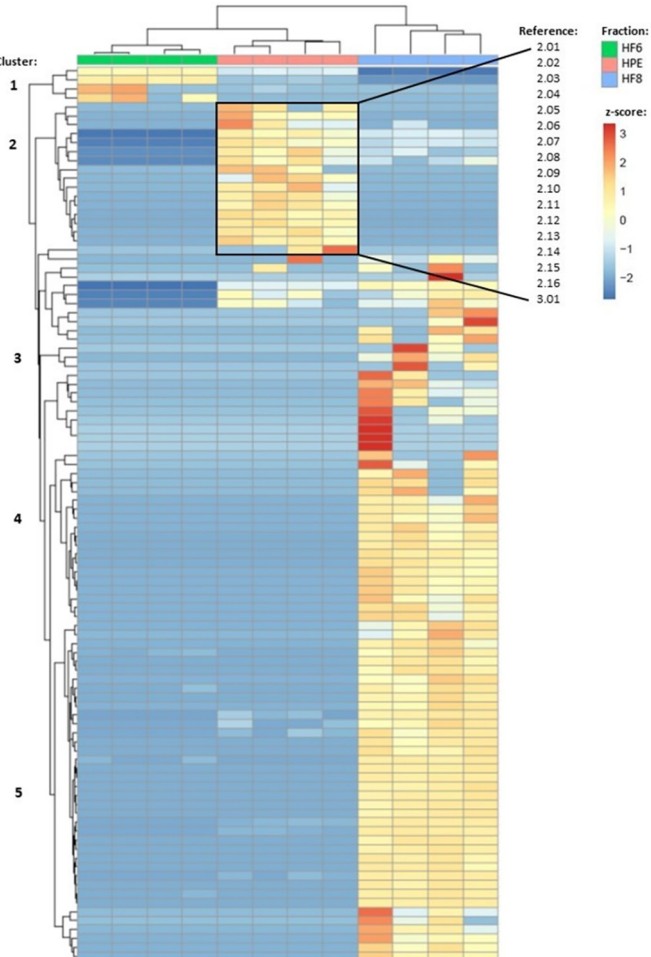

**Fig 2. Proteomic analysis of identified proteins in three HPLC-separated fractions of SRO hemolymph (HF6, HPE, HF8).** The simplified heatmap shows hierarchical clustering (Ward's method) of the quantitative values of the identified proteins in four replicates grouped based on scaled abundance in the respective fractions. Proteins with highest abundances in HPE (cluster 2) are listed in Table 1 under corresponding reference numbers. The heatmap with detailed protein annotations and complete proteomics data are provided in S2 Fig in S2 File.

20, 25, 37 and 60 kDa were most intense in HPE (S2 Fig in S2 File). A total of 100 proteins were identified (with 99% confidence) across HF6, HPE and HF8 with 7, 31, and 85 proteins in each fraction respectively (S1 Spreadsheets). Hierarchical clustering of proteomics data indicated unique clusters (Fig 2). Cluster 2 contained proteins with the highest abundance in HPE (Fig 2, S2 Fig in S2 File). Proteins in cluster 2 (and 3.01) and their functional annotations are listed in Table 1. There were 17 proteins these clusters of interest (Table 1, Fig 2). Of these, 12 proteins were unique to HPE (Fig 4). The others were also present in HF6 and/or HF8 but at lower abundances. Identified protein sequences shared strong similarity at the amino acid level to proteins from *S. glomerata*, other oyster species (*Crassostrea gigas, Crassostrea virginica*) and mussels (*Mytilus* spp.). There were no proteins that were unknown in HPE (S1 Spreadsheets).

HPE heat treatments caused increases in the intensity of SDS-PAGE bands around 20, 25, and 37 kDa indicating increased protein denaturation at the higher temperatures (S2 Fig in S2 File). Samples stored at -80˚C had the highest relative protein abundances (Fig 3; S1 Spreadsheets). Samples treated at 60˚C for 1 h showed lower abundances of 25 proteins, compared to

**Table 1. Proteins upregulated in HPE compared to HF6 and HF8, in order of appearance in heatmap cluster 2.** Cluster reference numbers correspond to those shown in Fig 3.

| Cluster reference | Identified Proteins | Accession no. | MW (kDa) | Uniqueness to HPE | Functional annotation (UniProt) | Evidence for direct antimicrobial activity | Refs |
|---|---|---|---|---|---|---|---|
| 2.01 | 40S ribosomal protein S28 (Fragment) OS = Arion vulgaris OX = 1028688 GN = ORF17713 PE = 3 SV = 1 | A0A0B6Y8M6_9EUPU (+9) | 9 | Yes | Calcium ion binding; muscle contraction | None | [55] |
| 2.02 | 60S ribosomal protein L11 OS = Mizuhopecten yessoensis OX = 6573 GN = KP79_PYT23786 PE = 3 SV = 1 | A0A210QZT5_MIZYE (+3) | 20 | Yes | Structural constituent of ribosome; protein translation | None | [55] |
| 2.03 | Cystatin B-like protein OS = Crassostrea gigas OX = 29159 GN = CGI_10013578 PE = 2 SV = 1 | D7EZH1_CRAGI | 11 | No; also present in HF8, but in much lower abundance | Cysteine-type endopeptidase inhibitor activity | Yes- cystatins show various antimicrobial properties | [56–66] |
| 2.04 | Tropomyosin OS = Saccostrea glomerata OX = 157728 PE = 2 SV = 1 | A0A2L1FDX2_9BIVA | 33 | No; also present in HF8, but in lower abundance | Structural constituent of ribosome; protein translation | None | [67, 68] |
| 2.05 | Carbonic anhydrase 1-like OS = Crassostrea virginica OX = 6565 GN = LOC111117514 PE = 3 SV = 1 | A0A8B8C9Q6_CRAVI | 35 | No; also present in HF8, but in lower abundance | Carbonate dehydratase activity, zinc ion binding; | Antimicrobial activity has been associated with bacterial carbonic anhydrase inhibitors, not the addition of carbonic anhydrases | [69, 70] |
| 2.06 | Tropomyosin OS = Crassostrea virginica OX = 6565 GN = LOC111130257 PE = 3 SV = 1 | A0A8B8DX08_CRAVI | 33 | No; also present in HF8, but in lower abundance | Plays a central role in the calcium dependent regulation of muscle contraction | None | [67, 68] |
| 2.07 | Peptidyl-prolyl cis-trans isomerase OS = Conus textile OX = 6494 PE = 2 SV = 1 | U5YDN8_CONTE | 17 | No; also present in HF8, but in lower abundance | Protein folding | None | [71] |
| 2.08 | 60S ribosomal protein L12 OS = Crassostrea gigas OX = 29159 GN = CGI_10000595 PE = 3 SV = 1 | K1PM66_CRAGI (+1) | 18 | Yes | Structural constituent of ribosome; protein translation | None | [55] |
| 2.09 | 40S ribosomal protein S25 (Fragment) OS = Arion vulgaris OX = 1028688 GN = ORF212606 PE = 3 SV = 1 | A0A0B7BUI7_9EUPU (+8) | 14 | Yes | Structural constituent of ribosome; protein translation | None | [55] |
| 2.10 | Catchin protein OS = Mytilus galloprovincialis OX = 29158 GN = catchin PE = 2 SV = 1 | Q9U0S5_MYTGA | 113 | Yes | Component of the myosin complex; muscle contraction | None | [72] |
| 2.11 | 40S ribosomal protein S4 OS = Mytilus coruscus OX = 42192 GN = MCOR_19036 PE = 3 SV = 1 | A0A6J8BHP0_MYTCO (+1) | 30 | Yes | Structural constituent of ribosome; protein translation | None | [55] |
| 2.12 | 60S ribosomal protein L6 OS = Crassostrea gigas OX = 29159 GN = CGI_10017767 PE = 3 SV = 1 | K1QW36_CRAGI | 26 | Yes | Structural constituent of ribosome; protein translation | None | [55] |
| 2.13 | 40S ribosomal protein S15 OS = Mytilus coruscus OX = 42192 GN = MCOR_21630 PE = 3 SV = 1 | A0A6J8BSV8_MYTCO (+5) | 17 | Yes | Structural constituent of ribosome; protein translation | None | [55] |

*(Continued)*

**Table 1.** (Continued)

| Cluster reference | Identified Proteins | Accession no. | MW (kDa) | Uniqueness to HPE | Functional annotation (UniProt) | Evidence for direct antimicrobial activity | Refs |
|---|---|---|---|---|---|---|---|
| 2.14 | 60S ribosomal protein L23a-like OS = *Crassostrea virginica* OX = 6565 GN = LOC111132903 PE = 3 SV = 1 | A0A8B8E7C9_CRAVI (+1) | 19 | Yes | Structural constituent of ribosome; protein translation | None | [55] |
| 2.15 | PDZ and LIM domain protein Zasp-like OS = *Crassostrea virginica* OX = 6565 GN = LOC111133459 PE = 4 SV = 1 | A0A8B8EDJ4_CRAVI | 18 | Yes | Metal binding; involved in muscle contraction and cytoskeleton organisation | None | [73] |
| 2.16 | Troponin T-like isoform X8 OS = *Crassostrea virginica* OX = 6565 GN = LOC111131563 PE = 3 SV = 1 | A0A8B8E358_CRAVI (+7) | 37 | Yes | Plays a central role in the calcium dependent regulation of muscle contraction | None | [74, 75] |
| 3.01 | Extracellular superoxide dismutase [Cu-Zn]-like OS = *Crassostrea virginica* OX = 6565 GN = LOC111113328 PE = 4 SV = 1 | A0A8B8BWJ1_CRAVI | 24 | Yes (but extracellular superoxide dismutase OS = Saccostrea glomerata OX = 157728 PE = 2 SV = 1 highly abundant in HPE, HF6 and HF8). | Catalyzes the dismutation of the superoxide radical into oxygen and hydrogen peroxide | Some evidence for antimicrobial activity. Plays a role in the inflammatory response *in vivo*. Pathogenic bacteria produce superoxide dismutases as virulence factors. | [76–81] |

other treatments (Fig 3). Of these, cystatin B-like protein and carbonic anhydrase 1-like protein were of interest for potential antibacterial activity (Figs 3 and 4, Table 1), whereas others were muscle contractile/regulatory proteins and ribosomal or transcription proteins (Figs 3 and 4, Table 1). There were two identified extracellular superoxide dismutase (eSOD) proteins: eSOD OS = *Saccostrea glomerata* OX = 157728 PE = 2 SV = 1 was abundant in all samples (HPE, HF6 and HF8; Fig 2 cluster 1; Fig 4), whilst eSOD [Cu-Zn] OS = *Crassostrea ariakensis* OX = 94323 PE = 2 SV = 1 was more abundant in HPE (Fig 2, cluster 3). Both identified eSODs were the only proteins that were *more* abundant in heat-treated (less active) HPE samples (Table 1, Figs 3 and 4, S1 Spreadsheets).

### 3.2 Antibacterial and antibiofilm activity of HPE and synergism with conventional antibiotics

**3.2.1 Antibacterial activity.** MBCs and median inhibitory concentrations ($EC_{50}$'s) for planktonic growth (i.e., killing of planktonic cells) are summarised in Table 2 and dose-response curves are shown in Figs 5–7. Antibiotic controls were effective within expected concentration ranges [82] in every assay providing data quality assurance (Table 2, S1 Table in S1 File).

*Individual HPE treatments.* In single-treatment assays, HPE was bactericidal to all *Streptococcus* spp. at low concentrations confirming the repeatability of previous results [30]. For reference, the effective concentration (MBC) of HPE against the *S. pneumoniae* ATCC strain was 8.03 μg/mL ($EC_{50}$ 3.69 μg/mL) (Table 2, Fig 5A). HF6 and HF8 were inactive (MBC and $EC_{50}$ values were not calculable). Clinical *S. pneumoniae* strains 14 and 19F were the most susceptible to HPE, with MBC's as low as 4.42 μg/mL and 4.82 μg/mL, respectively (Table 2, Fig 5B–5D). *S. pyogenes* was also susceptible to HPE with an MBC of 24.09 μg/mL (Table 2, Fig 6A). *S. aureus* (Gram positive) and all Gram-negative species of bacteria (*P. aeruginosa*, Nt*Hi*, *M.*

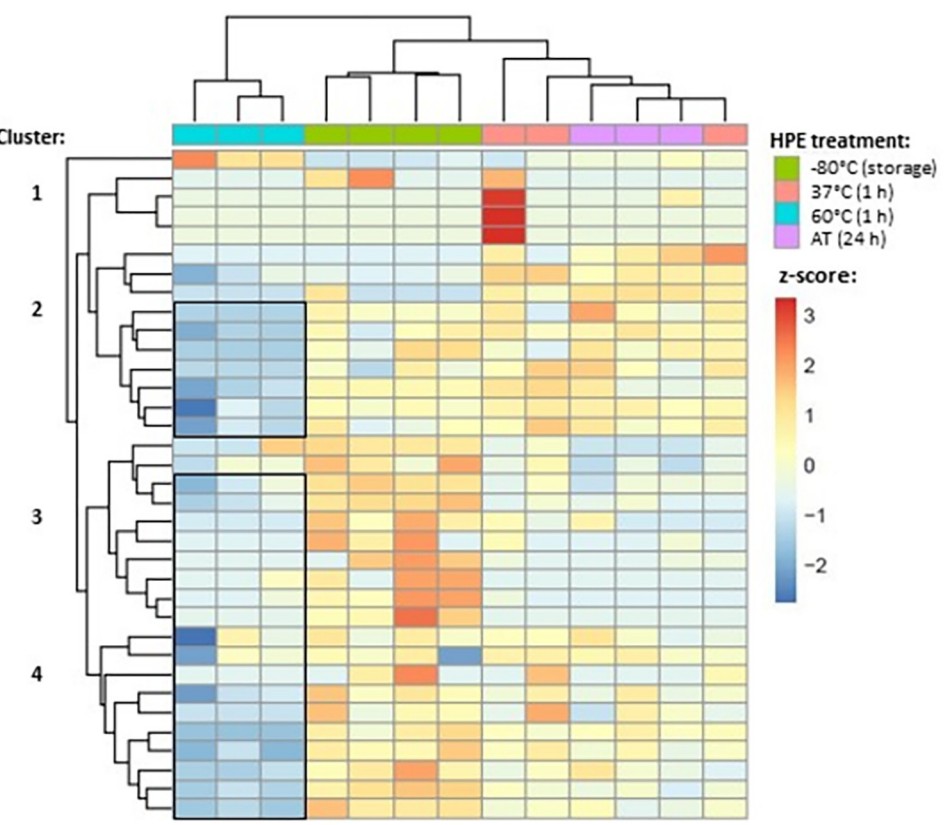

**Fig 3. Proteomics analysis of identified proteins in HPE maintained under different temperature conditions (storage at -80°C, storage at ambient temperature for 24 h, treated at 37°C for 1 h, treated at 60°C).** Proteins in clusters 2, 3 and 4 within black boxes had lower relative abundances in HPE treated at 60°C, correlating to a reduction in antimicrobial activity. The proteins are shown in Fig 4 (Venn diagram). The heatmap with detailed protein annotations and complete proteomics data are provided in S2 Fig in S2 File.

*catarrhalis* and *K. pneumoniae*) were not susceptible to HPE alone (Table 2, where MBCs were not calculable- NC; Figs 6 and 7).

*Combinations of HPE with antibiotics.* For all strains of *S. pneumoniae*, the presence of HPE at sub-MBC concentrations (1 µg/mL and 3 µg/mL) reduced the dose (i.e., improved the efficacy) of ampicillin required to inhibit growth (Fig 5). For the ATCC strain, MBCs for ampicillin were 0.25 µg/mL alone, 0.125 µg/mL when combined with 1 µg/mL HPE, and 0.06 µg/mL when combined with 3 µg/mL HPE (Table 2, Fig 5A). The effect of HPE-ampicillin combinations was most significant for clinical *S. pneumoniae* strains, especially strain 14 where there was a seven-fold reduction in the ampicillin MBC (from 0.125 µg/mL) with 1 µg/mL HPE (to 0.016 µg/mL), and a 30-fold reduction (to 0.004 µg/mL) with 3 µg/mL HPE (Table 2, Fig 5B–5D).

HPE at 3 µg/mL acted synergistically with ampicillin against *S. pyogenes*, halving the MBC for ampicillin from 0.125 µg/mL (alone) to 0.052 µg/mL (in combination) (Table 2, Fig 6A). HPE also acted synergistically to improve the activity of other conventional antibiotics against species that were not susceptible to HPE alone (Table 2, Figs 6 and 7). HPE at 12 µg/mL caused a marked five-fold reduction in the MBC of ampicillin against *S. aureus*, from 0.50 to 0.094 µg/mL (Table 2, Fig 6B). For *P. aeruginosa*, the MBC for gentamicin was 2.33 µg/mL, which was more than halved to 1.0 µg/mL in combination with 12 µg/mL HPE (Table 2, Fig 6C). For *M. catarrhalis*, the MBC for ciprofloxacin was 0.188 µg/mL, which was reduced four-fold to 0.047 µg/mL in combination with 12 µg/mL HPE (Table 2, Fig 7A). For *K. pneumoniae*, the

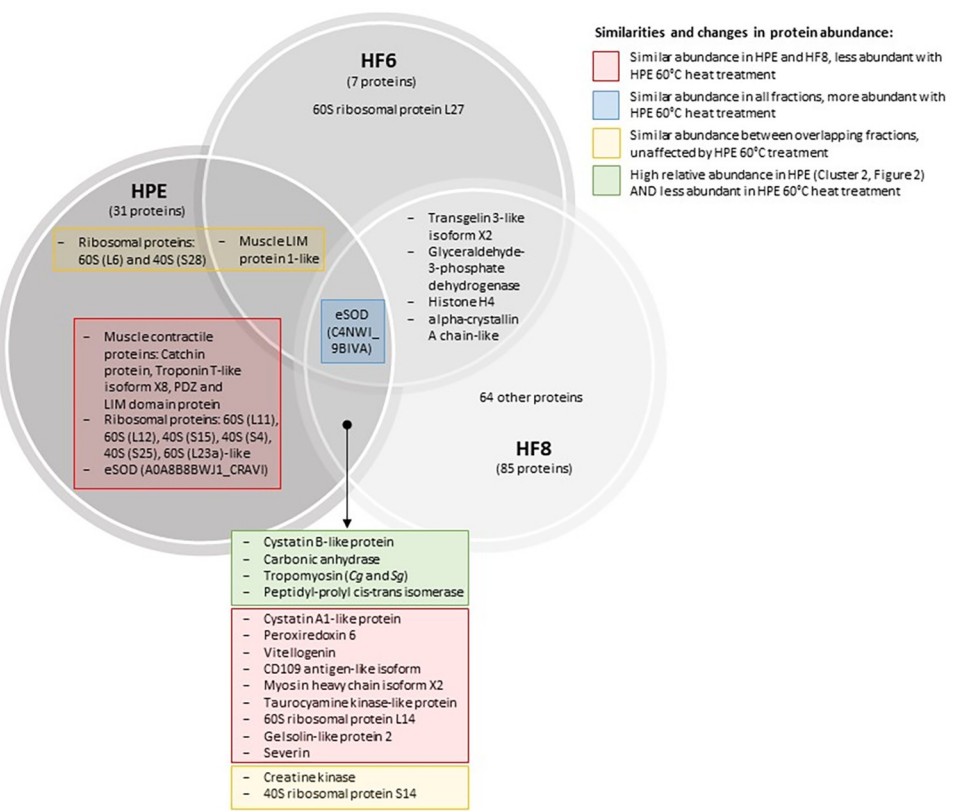

**Fig 4. Venn diagram of identified proteins to delineate the potential active proteins in HPE.** There were 17 proteins in cluster 2 (one in the first row of cluster 3) (Fig 2, Table 1), which showed the highest abundance in HPE. Of these, 12 proteins were unique to HPE. The active proteins must have been less abundant in HPE heated to 60˚C. The two Tropomyosin proteins in HPE and HF8 were Cg (*Crassostrea gigas*; A0A8B8DX08_CRAVI) and Sg (*Saccostrea glomerata*; A0A2L1FDX2_9BIVA).

MBC of 16 μg/mL trimethoprim was reduced eight-fold to 2 μg/mL in combination with 12 μg/mL HPE (Table 2, Fig 7B). For Nt*Hi*, the activity of ampicillin was only marginally improved in combination with 12 μg/mL HPE (Table 2, Fig 7C).

**3.2.2 Antibiofilm activity.** Biofilm inhibition data is provided as Figs 5, 6 and 8. Minimum biofilm inhibition concentrations and $EC_{50}$'s are provided as supplementary (S1 Table in S1 File). Biofilm inhibition concentrations for *Streptococcus* sp. were generally similar to antibacterial concentrations (because of the direct bactericidal effect on planktonic cells) (Figs 5 and 6, Table 2, S1 Table in S1 File). *S. pneumoniae* biofilm formation was completely inhibited by HPE at concentrations below MBC's, even for clinical strains 19F and 14 which formed thick biofilms (Fig 5B and 5D, left panels; S2 Fig in S2 File). In biofilm treatment assays, 9.63 μg/mL HPE was 100% bactericidal to *S. pneumoniae* (ATCC) cells embedded in pre-formed biofilms. The extracellular polymeric substance (EPS) matrix was not dissolved (i.e., remained adhered to plates, visible with the naked eye), but embedded cells were not viable (scraping and plating of remaining biofilms in wells treated with ≥9.63 μg/mL HPE yielded 0 CFU/mL). Formation of biofilms by *S. pyogenes*, *S. aureus* and *P. aeruginosa* were only weakly inhibited by individual HPE treatments (Fig 6, left panels) but, in combination assays HPE significantly improved the biofilm inhibition activity of conventional antibiotics (Figs 6 and 7, left panels). Biofilm inhibition data for other species was not calculable since they did not adhere to the plates even in positive growth control wells (Fig 8).

**Table 2. Effective antibacterial concentrations (µg/mL) of HPE and conventional antibiotics against a range of pathogens presented as MBC: Minimum bactericidal concentration (± standard deviations) and EC$_{50}$: Median effective concentration (with 95% highest posterior density intervals [HPDI]).** Effective concentrations are for antibiotics in the presence of HPE at sub-MIC concentrations (1–3 µg/mL) for susceptible species (*Streptococcus*) and at higher concentrations (6–12 µg/mL) for non/less-susceptible species (*S. aureus*, and Gram-negative sp.) NT: not tested, NC: not calculable.

| Species (strain) | Antibiotic | Effective antibacterial concentrations (µg/mL) MBC (±SD), EC$_{50}$ (95% HPDI's) | | | |
|---|---|---|---|---|---|
| | | HPE individual treatment | Antibiotic individual treatment | Antibiotic + 1 µg/mL HPE combination treatment | Antibiotic + 3 µg/mL HPE combination treatment |
| **Gram-positive** | | | | | |
| *S. pneumoniae* (ATCC 51916) | Ampicillin | **8.03** (±2.2), 3.65 (0.35, 4.65) | **0.25** (±0.0), 0.13 (0.09, 0.17) | **0.125** (±0.0), 0.08 (0.06, 0.11) | **0.06** (±0.0), 0.02 (0.00, 0.06) |
| *S. pneumoniae* (19F, clinical) | Ampicillin | **4.42** (±0.9), 2.09 (1.31, 3.07) | **0.078** (±0.03), 0.07 (0.01, 2.16) | **0.016** (±0.0), 0.01 (0.004, 0.01) | **0.008** (±0.0), 0.002 (0.001, 0.004) |
| *S. pneumoniae* (14, clinical) | Ampicillin | **4.82** (±0.0), 1.52 (0.99, 2.37) | **0.125** (±0.0), 0.04 (0.03, 0.06) | **0.016** (±0.0), 0.004 (0.002, 0.007) | **0.004** (±0.0), 0.0006 (0.0001, 0.0036) |
| *S. pneumoniae* (6B, clinical) | Ampicillin | **9.63** (±0.0), 2.61 (1.29, 4.97) | **0.06** (±0.0), 0.03 (0.02, 0.06) | **0.016** (±0.0), 0.008 (0.005, 0.010) | **0.008** (±0.0), 0.003 (0.002, 0.006) |
| *S. pyogenes* (ATCC 19615) | Ampicillin | **24.09** (±14.5), 7.09 (3.92, 13.01) | **0.125** (±0.0), 0.067 (0.045, 0.092) | NT | **0.052** (±0.015), 0.003 (0.001, 0.006) |
| | | | | Antibiotic + 6 µg/mL HPE combination treatment | Antibiotic + 12 µg/mL HPE combination treatment |
| *S. aureus* (ATCC 25923) | Ampicillin | NC, 209.9 (144.0, 457.6) | **0.50** (±0.0), 0.03 (0.01, 0.06) | NT | **0.094** (±0.03), 0.01 (0.004, 0.03) |
| **Gram-negative** | | | | | |
| Nt*Hi* (ATCC 10211) | Ampicillin | NC, 589.93* (248.39, 2760.04) | **0.29** (±0.093), 0.050 (0.018, 0.13) | NT | **0.25** (±0.0), 0.098 (0.057, 0.16) |
| *M. catarrhalis* (ATCC K65) | Ciprofloxacin | NC, 530.60* (254.42, 1081.39) | **0.19** (±0.06), 0.027 (0.011, 0.076) | NT | **0.047** (±0.01), 0.0035 (0.001, 0.0082) |
| *K. pneumoniae* (ATCC 51916) | Trimethoprim | NC, 229.29* (94.54, 749.95) | **16.0** (±0.0), 5.84 (2.69, 11.63) | NT | **2.0** (±0.0), 0.19 (0.03, 1.27) |
| *P. aeruginosa* (385, clinical) | Gentamicin | NC, 235.57* (157.75, 678.58) | **2.33** (±0.74), 1.46 (1.02, 2.01) | **2.0** (±0.0), 0.65 (0.53, 0.82) | **1.0** (±0.0), 0.55 (0.44, 0.73) |

*Less reliable as estimated EC$_{50}$ concentrations are greater than the maximum tested concentration (150 µg/mL).

The activity of HPE was strongest when maintained at -80˚C (MBC 7.23 µg/mL). Antibacterial activity was significantly lower when HPE was stored at ambient laboratory temperature for 24 h (MBC 9.63 µg/mL) and when heated to 37˚C (MBC 9.63 µg/mL) (p<0.05). Reduction in HPE activity with treatment at 60˚C was most marked (MBC 19.27 µg/mL) compared to all other samples (p<0.05) (Fig 8).

### 3.3 Cytotoxicity

HPE showed no cytotoxicity toward healthy human lung cells (A549) within the tested concentration range (0.09–205 µg/mL), which was inclusive of effective antibacterial doses (i.e., 1–24 µg/mL) (Fig 9). Measurements of cell viability at all treatment levels, even at the upper limit of 205.0 µg/mL HPE, were not significantly different to the media-only control (Fig 9, p > 0.05). CC$_{50}$ values were therefore not calculable.

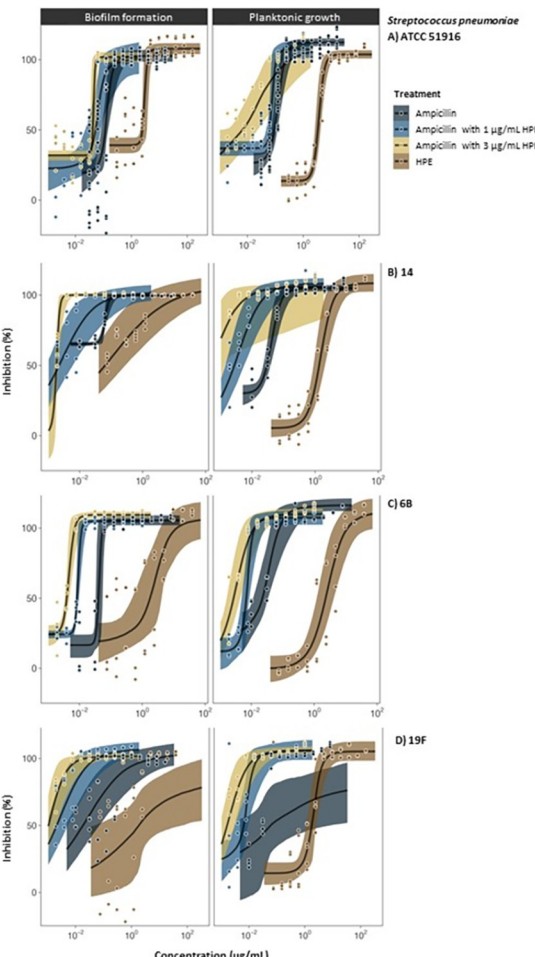

**Fig 5. Antibacterial (inhibition of planktonic growth) and antibiofilm (inhibition of biofilm formation) activity of SRO hemolymph fraction HPE alone, conventional antibiotics (ampicillin) alone, and HPE-antibiotic combination treatments where HPE was combined at fixed concentrations (1 or 3 μg/mL) with ampicillin at variable concentrations (1.0–0.002 μg/mL) tested against laboratory (ATCC 51916) (A) and clinical (14, 6B, 19F) strains (B-D) of *S. pneumoniae*.**

## 4. Discussion

Our work has provided evidence that a semi-purified protein fraction of SRO hemolymph (HPE) has *in vitro* antibacterial properties against a range of clinically important respiratory bacteria, and increases the efficacy of antibiotics used as conventional treatments. This provides a foundation for the development of HPE and constituent AMPPs as novel antibiotics. HPE showed a high degree of selectivity toward *Streptococcus* spp. (Fig 5, Table 2). Other typically resistant/Gram-negative bacteria were unaffected by HPE in single-treatment assays, but HPE acted synergistically with conventional antibiotics used against them (Figs 6, 7, Table 2).

The potential utility of HPE as a novel treatment for *Streptococcus* spp. infections is notable. *S. pneumoniae* strains used in this study are covered by available pneumococcal conjugate vaccines (PCVs) (PCV-7, -13, -15 and -20) and the pneumococcal polysaccharide vaccine (PPV-23). However, non-vaccine and drug-resistant serotypes persist and may be susceptible to HPE [83]. There is no available *S. pyogenes* vaccine so antibiotics remain broadly essential. HPE should therefore be tested against non-vaccine and drug-resistant *Streptococcus* spp. strains.

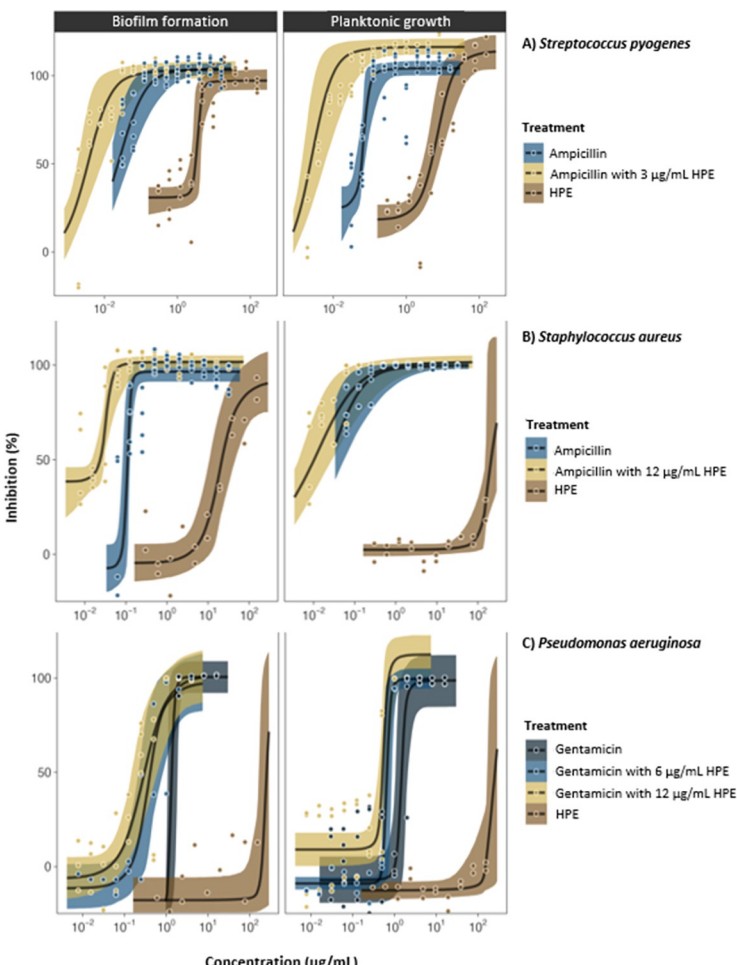

**Fig 6.** Antibacterial (inhibition of planktonic growth) and antibiofilm (inhibition of biofilm formation) activity of SRO hemolymph fraction HPE alone, conventional antibiotics alone, and HPE-antibiotic combination treatments where HPE was combined at fixed concentrations (3, 6 or 12 μg/mL) with conventional antibiotics (ampicillin and gentamicin) at variable concentrations (1.0–0.002 μg/mL and 4.0–0.008 μg/mL, respectively) tested against laboratory strains of *S. pyogenes* (A) and *S. aureus* (B), and a clinical strain of *P. aeruginosa* (C).

The strong bactericidal activity of HPE toward the streptococcus, complementary activity toward other species in HPE-antibiotic combinations, and lack of/minimal activity in both single and combination treatments toward Nt*Hi*, may offer insight into the HPE mechanism of action.

AMP-antibiotic combination/adjuvant therapies have been widely tested and recommended in recent years to improve the performance of available drugs [84–88]. Our results are in agreement since HPE at sub-MBC concentrations (1.0–12.0 μg/mL) improved the efficacy of ampicillin, ciprofloxacin, gentamicin and trimethoprim between 2 and 32-fold. HPE may therefore be especially useful in combination therapies to revive the use of available antibiotics against drug-resistant bacterial species/strains. Combination results for *S. aureus* and *P. aeruginosa* are particularly relevant as these species are renowned for high levels of resistance; there are examples of other AMPP-antibiotic combinations at similar effective concentrations proving useful against these species [89–94]. Interference with the integrity/permeability of the bacterial cell membranes caused by AMPPs is a mechanism by which antibiotics can reach targets more easily [11]. In synergistic combinations with conventional antibiotics, the active

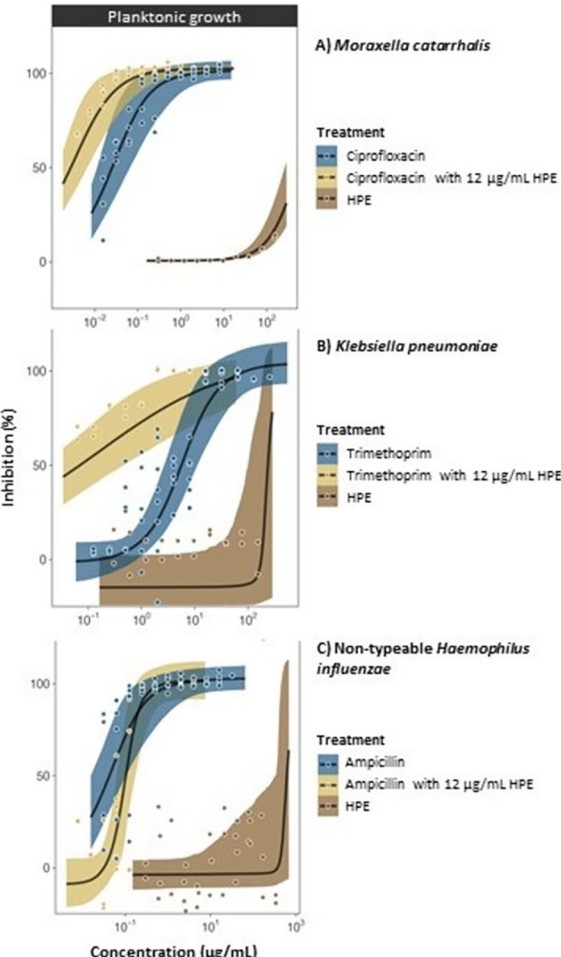

**Fig 7. Antibacterial (inhibition of planktonic growth) activity of SRO hemolymph fraction HPE alone, conventional antibiotics alone, and HPE-antibiotic combination treatments where HPE was combined at fixed concentrations (12 µg/mL) with conventional antibiotics (ciprofloxacin, trimethoprim and ampicillin) at variable concentrations (8.0–0.015 µg/mL, 32.0–0.06 µg/mL and 4.0–0.008 µg/mL, respectively) tested against a clinical strain of *M. catarrhalis* (A) and laboratory strains of *K. pneumoniae* (B) and Nt*Hi* (C).**

components of HPE must convey independent (e.g., direct enzyme inhibition) and complementary (e.g., impaired membrane integrity) mechanisms to disrupt cellular function.

AMPPs frequently fail to reach market because of high production costs and intermediate efficacy (active in the micromolar range) compared to conventional antibiotics (active in the nanomolar range) [18]. In this study, we used HPE-antibiotic combinations to minimise required doses of antibiotics *and* HPE (i.e., HPE was active at 4.42–24.09 µg/mL alone but was active at 1.0–3.0 µg/mL in combination assays against *Streptococcus* spp., and was not active alone against Gram negative species or *S. aureus* but combined with conventional antibiotics was active at 6.0–12.0 µg/mL). Combination approaches therefore may not only revive the use of available drugs but reduce the required doses of HPE such that scales of production would be feasible.

Aside from direct antimicrobial activity, many AMPPs are able to modulate the host immune response (e.g., by increasing chemokine production, enhancing wound healing and angiogenesis, exerting pro- and anti-apoptotic effects on different immune cell types, as well as having adjuvant activity to promote adaptive immunity) [95, 96]. Future research investigating

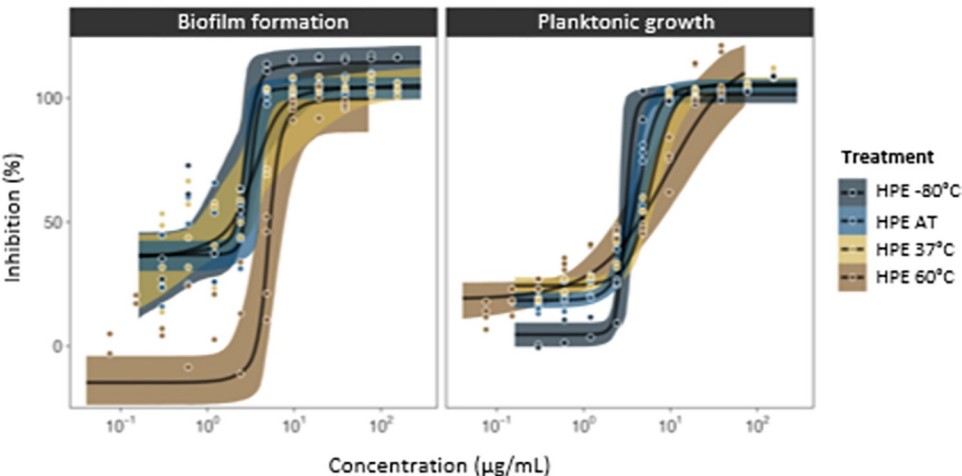

**Fig 8. Reduction in antibacterial and antibiofilm activity of SRO hemolymph fraction HPE following different heat treatments (storage at -80˚C, storage at ambient laboratory temperature for 24 h [AT], heated to 37˚C for 1 h or heated to 60˚C for 1 h) tested against *Streptococcus pneumoniae* (ATCC 51916) at concentrations between 154.15–0.30 μg/mL.**

immune-modulatory activities of HPE/constituent AMPPs is warranted to support its potential adjuvant properties in therapeutic combinations.

Bacteria in pre-formed biofilms can typically withstand much higher (up to 1000 times) concentrations of antibiotic treatments [3]. Here, *S. pneumoniae* (ATCC 51916) cells in pre-formed biofilms were killed by HPE at 9.63 μg/mL, which is only slightly higher than the MBC for planktonic cells (8.03 μg/mL) and concentrations inhibiting biofilm formation were the same or lower than MBCs (Table 1, S1 Table in S1 File). This suggests that the active constituent/s of HPE can readily penetrate/affect cells embedded within the *S. pneumoniae* EPS matrix, which many antibiotics cannot. For example, Vandevelde et al. [97] tested different classes of antibiotics, including β-lactams (amoxicillin), macrolides (clarithromycin, solithroycin), and fluoroquinolones (levofloxin, moxifloxin), against similar 2-day old *S. pneumoniae* biofilms. To achieve 50% reductions in viability of *S. pneumoniae* in biofilms, concentrations of antibiotics were between 0.1 and 111 times the MIC values for planktonic growth [97].

The antibacterial activity of HPE is strong relative to other tested AMPPs and hemolymph protein extracts. Effective concentrations of patented AMPPs range from 1–100 μg/mL, most

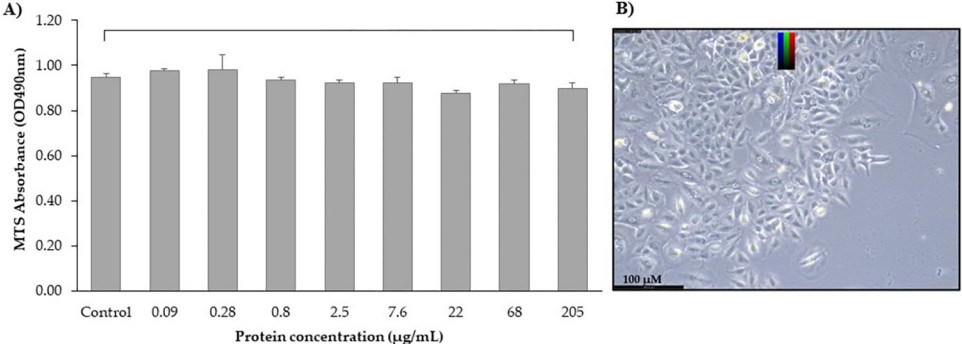

**Fig 9. Effect of HPE on the viability of A549 human lung cells measured by the MTS assay.** A) There was no significant difference in cell viability (MTS absorbance, which is inversely proportional to viable cell activity) with HPE treatments between 0.09–205 μg/mL protein and controls (no HPE added). B) Image of healthy cells exposed to HPE.

being around 25–50 μg/mL [98, 99]. All FDA-approved AMPPs (Colistin [Polymyxin E], Polymyxin B, Nisin, Melittin, and Daptomycin) are active at similar concentrations to HPE against Gram-positive species: MICs are between 0.25–8.0 μg/mL for Colistin [100], 2.0–8.0 μg/mL for Polymyxin B [101, 102], 1.0–83.0 μg/mL for Nisin [103], 1.0–32.0 for Melittin [104, 105] (but is cytotoxic >6.5 μg/mL [105]), and 0.125–8.0 μg/mL for Daptomycin [106, 107]. Other AMPPs currently in the clinical trials pipeline are effective at similar concentrations, for example MICs of 16–64 μg/mL Pexiganan [108], 6.25–100 μg/mL human lactoferrin HLF1-11 [109], 1–8 g/mL Brilacidin [110], 1–1024 μg/mL Omignan [111], 250–4000 μg/mL NP213 [Novotaxin®] [112]. The antimicrobial activity of HPE is also similar to some of the most promising molluscan AMPP leads, including defensins; more detailed comparisons between HPE and other molluscan AMPPs are provided in S1 Table in S1 File and in our previous work [30]. Further purification may increase the effectiveness of HPE; a caveat to this is that the combinations of proteins and peptides comprising the extract may be important, as is common among AMPPs, and further purification steps risk degrading the active components.

The proteins responsible for observed activity must have showed: the highest relative abundance in HPE, lower but at least some abundance in HF8 (some activity), and lowest or no abundance in HF6 (lowest activity) (Fig 4). The active proteins in HPE must have also been more abundant in HPE maintained at -80°C (strongest activity) compared to HPE treated at 60°C (reduced activity). Activity can therefore be ascribed to cystatin B-like protein, carbonic anhydrase, tropomyosin, and/or peptidyl-prolyl cis-trans isomerase (Fig 4, S1 Table in S1 File).

Cystatins were the most abundant known AMPPs present in HPE, in this study and in our previous work (Table 1, S1 Table in S1 File). Cystatins (cysteine protease inhibitors) are important effectors of immunity/defense in humans [56], animals [113, 114] and plants [115, 116], and have been extensively studied. The activity of cystatins from different sources tested against various human and environmental pathogens ranges from 16 to 200 μg/mL (S1 Table in S1 File). Effective concentrations reported for other cystatins suggest that the activity of cystatin B-like protein in HPE is relatively strong if it is indeed the active AMPP. To our knowledge, cystatins have not been previously tested in combination with antibiotics. Isolation of cystatin- B-like protein from HPE may be possible by size-exclusion chromatography (e.g., cystatins 11–14 kDa vs. carbonic anhydrase 35 kDa) and affinity chromatography, since cystatins show binding affinity toward papain [117], followed by SDS-PAGE to confirm purity (S1 Table in S1 File).

Carbonic anhydrase was also highly abundant in HPE. *Inhibiting* carbonic anhydrases is a recognised antimicrobial strategy [118]. Conversely, adding or *promoting* carbonic anhydrase activity lowers environmental pH which may be advantageous [70, 119], for example the activity of cystatins is improved under mildly acidic conditions (pH 5.0–6.5) [119]. Other candidate proteins in HPE including tropomyosin and peptidyl-prolyl cis-trans isomerase have no known antimicrobial activity or functional units. Ultimately, more research is needed to understand the constituent AMPPs and mechanism of action.

Peptides/proteins present unique opportunities and challenges for the pharmaceutical industry compared to small molecules. The high solubility and stability of HPE in media are favorable properties, and the reproducibility of the method to obtain this fraction (as demonstrated in by this study and [30]) lend favorably to therapeutics registration. Data for HPE treated at laboratory temperature (i.e., the administration environment) and 37°C (i.e., human body temperature 37°C) are environmentally relevant and desirable for storage and handling, but prolonged exposure to temperatures above freezing and repeated freeze-thaw cycles should be avoided to minimise protein denaturation and potential losses in activity. It is important to note that lower chemical stability is sometimes desirable. For example, if AMPPs are used in

an adjuvant capacity, the precise immunomodulatory effect desired may depend on a limited half-life (i.e., moderate, but not overstimulate immunity). The natural degradation of AMPPs also makes their environmental fate less problematic than many conventional antibiotics.

Cytotoxicity toward healthy human cells can compromise the therapeutic potential of some AMPs [18]. In this study, the safety profile of HPE, specifically for respiratory applications, was demonstrated by lack of toxicity to the A549 human lung cell line at concentrations far exceeding effective antimicrobial concentrations. It may also be suitable for topical, inhalation, injection, and surface applications, but would likely be unstable in the gastrointestinal tract. The safety of HPE is demonstrated more generally by the extensive use of oysters as traditional medicines, as a food source, and as functional foods [29].

There are strict regulations and requirements for clinical testing, quality control and registration of all health-care products, even those from natural origins, which depend on the product classification and authority [120, 121]. HPE holds potential for registration as a nutraceutical extract or pharmaceutical-grade AMPP if further purified. In either case, substantial research will be required to demonstrate quality, safety, efficacy and progress through the drug development pipeline (Fig 10). Developing a high-value medicinal product from HPE

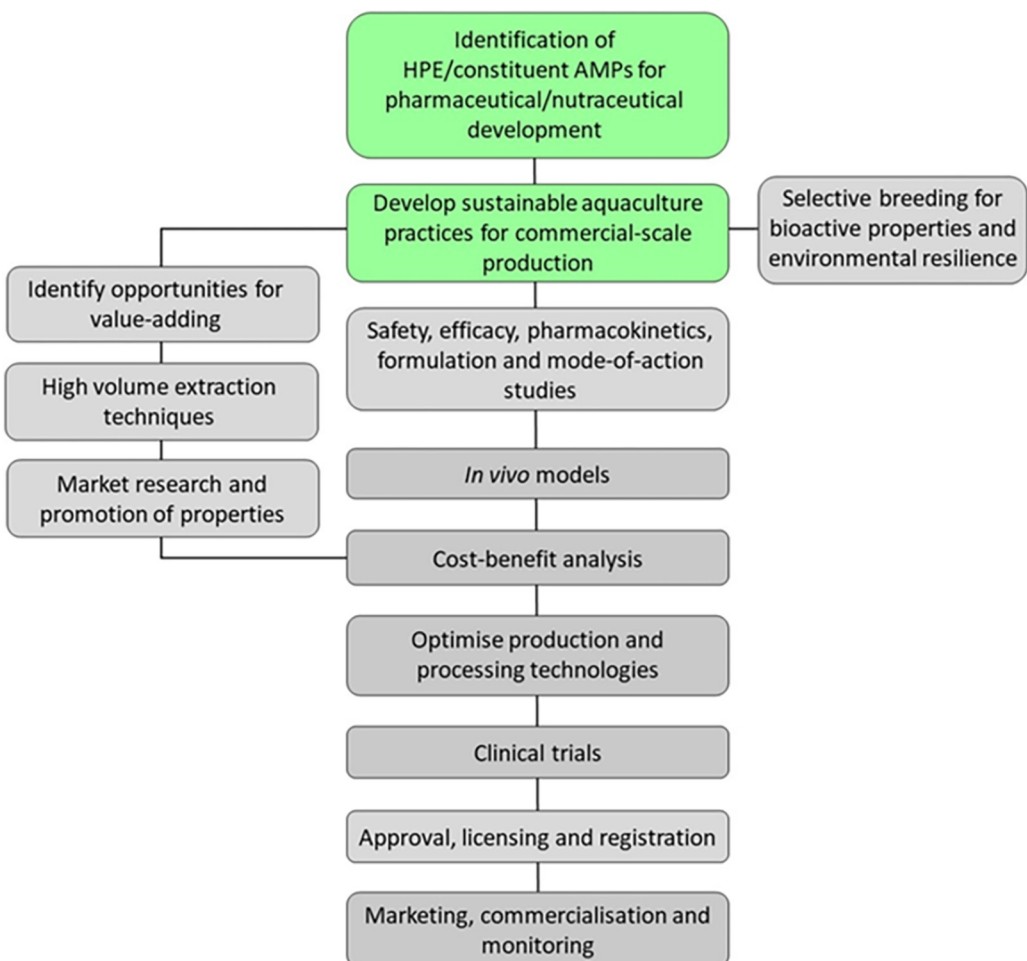

**Fig 10. Major research and development steps required for development of a therapeutic product from HPE/constituent AMPPs.** Boxes in green are well-developed. Boxes in grey are yet to be addressed in full.

will therefore require strategic planning and substantial financial investment, but markets for medical and healthcare products are robust and ever growing [122]. For example, the global marine-based drug market is currently estimated at US$ 4,177.9 million [123]; sales of peptide-based drugs exceeded US$ 70 billion in 2019 [124], and the nutraceutical market was valued at US$ 561 billion in 2022 with a growth rate of over 5% each year [125]. Investing in the development of HPE is risky, but if successful, returns could be very high.

Extensive harvest of natural products from wild populations of marine organisms is not often economical or ecologically possible [121]. In this case, SRO represent the focus of commercial aquaculture production in Australia so are already produced at commercial scales [126]. Consequently, sustainable SRO harvest and HPE supply is possible and presents a great opportunity for the traditionally low-tech aquaculture industry, at least until chemical synthesis is successful and commercially-viable. In a review of marine biotechnology development challenges and market trends, Daniotti and Re [127] found that projects focused on optimising cultivation conditions, harvesting and extraction methods showed high potential for market readiness and business development, compared to projects focused on the drug discovery challenge (e.g., omics, bioinformatics, and pharmacological analysis). Practical opportunities for the aquaculture sector may therefore include: expanding occupation of available oyster leases, increasing the value of "seconds" (i.e., smaller, misshaped SRO), allowing harvest during periods where water quality does not comply with food safety regulations [128], but may be acceptable for processing HPE, and developing methods for high-volume hemolymph extraction.

Investigation into the influence of oyster condition and sampling time (i.e., climate, water quality and organism lifecycle) on HPE bioactivity is warranted as these factors may influence variations in effective concentrations between studies. Further research on HPE should be undertaken using SRO collected during their peak season (Spring to Autumn), but not during or directly after spawning, and when catchments have been unaffected by significant weather or contamination events in the three months leading up to collection.

## 5. Conclusion

New antibiotics are needed and AMPPs from marine invertebrates are targeted as productive leads. HPE, a semi-purified extract of SRO hemolymph proteins, showed strong antibacterial activity against *Streptococcus* spp. Combination assays using 1–12 μg/mL HPE with conventional antibiotics significantly improved the efficacy of antibiotic treatments against a range of other respiratory pathogens *in vitro*. In practice, this could both reduce overexposures to available antibiotics and reduce the barriers to further development and clinical implementation of HPE/active AMPPs. HPE is also non-toxic with good chemical stability, which further supports the viability of constituent AMPPs as drug candidates. Further research and investment to progress HPE through the drug development pipeline is worthwhile and represents a great opportunity for collaboration between researchers, regulators and industry.

## Supporting information

**S1 File.**
(DOCX)

**S2 File.**
(DOCX)

**S1 Spreadsheets.**
(XLSX)

## Acknowledgments

The authors wish to gratefully acknowledge Dr. Matthijs Hollanders for assistance with statistics, Dr. Jessica Browne for supplying the clinical isolates, and Dr. Andrea Bugarcic for supplying the A549 cells.

## Author Contributions

**Conceptualization:** Kate Summer, Kirsten Benkendorff.

**Data curation:** Kate Summer, Sarah Giles.

**Formal analysis:** Kate Summer, Qi Guo.

**Funding acquisition:** Kate Summer.

**Investigation:** Kate Summer.

**Methodology:** Kate Summer, Lei Liu, Bronwyn Barkla, Sarah Giles.

**Project administration:** Kate Summer.

**Resources:** Bronwyn Barkla, Kirsten Benkendorff.

**Supervision:** Bronwyn Barkla.

**Visualization:** Kate Summer, Qi Guo.

**Writing – original draft:** Kate Summer.

**Writing – review & editing:** Kate Summer, Qi Guo, Lei Liu, Bronwyn Barkla, Sarah Giles, Kirsten Benkendorff.

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
