## [Decision Letter · Decision Letter 0]

14 Jun 2024

PONE-D-24-09897Antimicrobial proteins from oyster hemolymph improve the efficacy of conventional antibioticsPLOS ONE

Dear Dr. Summer,

Thank you for submitting your manuscript to PLOS ONE. After careful consideration, we feel that it has merit but does not fully meet PLOS ONE’s publication criteria as it currently stands. Therefore, we invite you to submit a revised version of the manuscript that addresses the points raised during the review process.

We look forward to receiving your revised manuscript.

Kind regards,

Maria del Mar Ortega-Villaizan

Academic Editor

PLOS ONE

Journal Requirements:

Additional Editor Comments:

The manuscript and work carried out by authors it is of interest since it addresses a critical issue in antimicrobial therapy and presents promising data on the use of oyster hemolymph proteins.

However, there are some points that should be clarified and improved, and additional information should be added. Please follow reviewer indications.

Reviewers' comments:

Reviewer's Responses to Questions

**Comments to the Author**

1. Is the manuscript technically sound, and do the data support the conclusions?

Reviewer #1: Partly

Reviewer #2: Yes

Reviewer #3: Yes

2. Has the statistical analysis been performed appropriately and rigorously? 

Reviewer #1: Yes

Reviewer #2: Yes

Reviewer #3: Yes

3. Have the authors made all data underlying the findings in their manuscript fully available?

Reviewer #1: No

Reviewer #2: Yes

Reviewer #3: Yes

4. Is the manuscript presented in an intelligible fashion and written in standard English?

Reviewer #1: Yes

Reviewer #2: Yes

Reviewer #3: Yes

5. Review Comments to the Author

Reviewer #1: Overall, this manuscript addresses a critical issue in antimicrobial therapy and presents promising data on the use of oyster hemolymph proteins. With detailed methodologies and a balanced discussion, it has the potential to make a significant impact in the field of antimicrobial research. This manuscript provides a comprehensive and detailed description of covering the preparation and treatment of hemolymph protein extracts (HPE), proteomic analyses, antibacterial and antibiofilm assays, and cytotoxicity assays. However, there are some areas where clarity and conciseness could be improved, and additional information could enhance reproducibility.

Specific Comments

2.1 HPE Preparation

Hemolymph Collection from SRO (Section 2.1.1):

Ensure that the referenced method [36] is accessible and sufficiently detailed for replication.

The statement "Hemolymph was withdrawn from multiple oysters, combined to obtain 5 mL pools" would benefit from specifying how many oysters were used on average per pool, as this could impact the consistency and yield of hemolymph.

Hemolymph Fractionation (Section 2.1.2):

The freeze-drying process is well-explained, but adding details about the reconstitution process (e.g., the volume of Milli-Q water used) would improve clarity.

HPE Heat Treatments (Section 2.1.3):

The heat treatment conditions are well-documented. It would be beneficial to include a brief rationale for the chosen temperatures and durations to understand their relevance to the study's objectives.

2.2 Proteomics

Protein Quantification and Visualization (Section 2.2.1):

Ensure that the sources of all reagents and materials are consistently specified (e.g., "Bio-Rad, Australia" for the protein dye reagent).

Protein Identification by HPLC-MS/MS (Section 2.2.2):

Mentioning the specific version of the database used for searching spectra (e.g., the release date or version number of the Uniprot Mollusca database) would enhance reproducibility.

The protein extraction and digestion process is well-described, but clarifying the final volume in which peptides are redissolved before analysis could prevent confusion.

Protein Data Analysis (Section 2.2.3):

Including an explanation for choosing NSAF and the specific clustering methods (PCA and hierarchical clustering) would provide context for readers unfamiliar with these techniques.

2.3 Antibacterial and Antibiofilm Assays

Media and Reagents (Section 2.3.1):

Ensure that any deviations from standard protocols (if any) are mentioned.

Bacteria Preparation (Section 2.3.2):

The preparation of bacterial cultures is detailed and aligns with standard procedures. Specifying the rationale for selecting the particular strains and clinical isolates would add context.

Antibacterial-biofilm Inhibition Coupled Assays (Section 2.3.3):

The assay setup and conditions are well-documented. The choice of concentrations for HPE and antibiotics could be briefly justified to highlight their relevance to the study.

The statistical methods used for data analysis are appropriate. Consider providing additional details on the criteria for determining significant differences.

Biofilm Treatment Assays (Section 2.3.4):

It would be helpful to explain why specific controls were chosen (e.g., EDTA and ciprofloxacin) to contextualize their relevance as benchmarks.

2.4 Cytotoxicity Assays

Cell Viability Using the MTS Assay (Section 2.4.3):

Justifying the range of HPE concentrations tested would add value, as would a brief discussion on the selection of statistical methods for data analysis.

The discussion is comprehensive and aligns well with the study's findings and objectives. Enhancing the structure, providing more detailed comparisons with existing research, and elaborating on future research directions will strengthen the manuscript.

Antibacterial Efficacy: Provide detailed comparisons with other AMPs.

Synergistic Effects: Expand on mechanisms and cite relevant studies.

Clinical Relevance: Discuss potential integration into clinical protocols.

Challenges: Elaborate on production costs and efficacy barriers.

Future Research: Specify next steps and potential collaborative efforts.

Reviewer #2: Authors did interesting research on Antimicrobial proteins from oyster hemolymph improve the

3 efficacy of conventional antibiotics and they found that HPE is a semi-purified extract of SRO hemolymph proteins showing strong antibacterial activity specific to Streptococcus sp. which could be used as potent agent to develop drug against streptococcus infections. I highly recommended the manuscript and have a future scope. Some grammar and spelling mistakes should be removed in revised thesis.

Reviewer #3: I have reviewed the manuscript entitled “Antimicrobial proteins from oyster hemolymph improve the efficacy of conventional antibiotics” submitted for possible publication in the journal “PLOS ONE”. The authors have done good efforts in compiling the results and all of the other relevant information. The work is thorough and the authors have described it very well. The manuscript could be beneficial for the reader and scientific community specially in terms of antibiotic resistance. After critically reviewing the manuscript, I did not noticed any flaws of major problems. However, few grammatical corrections have to be considered before proceeding it further towards publication process. My specific comments are:

1. Line 154: Correct the sign “°C”.

2. Line 155: correct “up to 30 50ms”

3. Figure 1 should be improved. The authors can use simply PPT to make the labelling in the figure.

6. PLOS authors have the option to publish the peer review history of their article (what does this mean?). If published, this will include your full peer review and any attached files.

Reviewer #1: No

Reviewer #2: **Yes: **Saiqa Andleeb

Reviewer #3: **Yes: **Naveed Ahmed

---

## [Author Response · Author response to Decision Letter 0]

12 Aug 2024

Please see response to reviewers attached.

---

## [Decision Letter · Decision Letter 1]

12 Sep 2024

Antimicrobial proteins from oyster hemolymph improve the efficacy of conventional antibiotics

PONE-D-24-09897R1

Dear Dr. Summer,

We’re pleased to inform you that your manuscript has been judged scientifically suitable for publication and will be formally accepted for publication once it meets all outstanding technical requirements.

Kind regards,

Maria del Mar Ortega-Villaizan

Academic Editor

PLOS ONE

Additional Editor Comments (optional):

Reviewers' comments:

Reviewer's Responses to Questions

**Comments to the Author**

1. If the authors have adequately addressed your comments raised in a previous round of review and you feel that this manuscript is now acceptable for publication, you may indicate that here to bypass the “Comments to the Author” section, enter your conflict of interest statement in the “Confidential to Editor” section, and submit your "Accept" recommendation.

Reviewer #3: All comments have been addressed

2. Is the manuscript technically sound, and do the data support the conclusions?

Reviewer #3: Yes

3. Has the statistical analysis been performed appropriately and rigorously? 

Reviewer #3: Yes

4. Have the authors made all data underlying the findings in their manuscript fully available?

Reviewer #3: Yes

5. Is the manuscript presented in an intelligible fashion and written in standard English?

Reviewer #3: Yes

6. Review Comments to the Author

Reviewer #3: (No Response)

7. PLOS authors have the option to publish the peer review history of their article (what does this mean?). If published, this will include your full peer review and any attached files.

Reviewer #3: No

---

## [Editor Report · Acceptance letter]

17 Oct 2024

PONE-D-24-09897R1 

PLOS ONE

Dear Dr. Summer, 

I'm pleased to inform you that your manuscript has been deemed suitable for publication in PLOS ONE. Congratulations! Your manuscript is now being handed over to our production team.

Kind regards, 

on behalf of

Dr. Maria del Mar Ortega-Villaizan 

Academic Editor

PLOS ONE